# Physicochemical Properties of Nanoencapsulated Essential Oils: Optimizing D-Limonene Preservation

**DOI:** 10.3390/polym17030348

**Published:** 2025-01-27

**Authors:** Diner Mori-Mestanza, Iraida Valqui-Rojas, Aline C. Caetano, Carlos Culqui-Arce, Rosita Cruz-Lacerna, Ilse S. Cayo-Colca, Efraín M. Castro-Alayo, César R. Balcázar-Zumaeta

**Affiliations:** 1Instituto de Investigación, Innovación y Desarrollo para el Sector Agrario y Agroindustrial (IIDAA), Universidad Nacional Toribio Rodríguez de Mendoza de Amazonas, Chachapoyas 01001, Peru; diner.mori@untrm.edu.pe (D.M.-M.); 7369695081@untrm.edu.pe (I.V.-R.); carlos.culqui@untrm.edu.pe (C.C.-A.); rosita.cruz@untrm.edu.pe (R.C.-L.); efrain.castro@untrm.edu.pe (E.M.C.-A.); 2Instituto de Investigación para el Desarrollo Sustentable de Ceja de Selva, Universidad Nacional Toribio Rodríguez de Mendoza de Amazonas, Calle Universitaria N° 304, Chachapoyas 01001, Peru; aline.caetano@untrm.edu.pe; 3Facultad de Ingeniería Zootecnista, Agronegocios y Biotecnología, Universidad Nacional Toribio Rodríguez de Mendoza de Amazonas, Chachapoyas 01001, Peru; icayo.fizab@untrm.edu.pe

**Keywords:** citrus, gas chromatography, stability, antioxidants

## Abstract

Essential oils exhibit antioxidant properties but are prone to oxidative degradation under environmental conditions, making their preservation crucial. Therefore, the purpose of this work was to evaluate the physicochemical properties of nanoencapsulated essential oils (EOs) extracted from the peel of sweet lemon, mandarin, lime, and orange using four formulations of wall materials consisting of gum arabic (GA), maltodextrin (MD), and casein (CAS). The results showed that EOs from sweet lemon, mandarin, lime, and orange showed higher solubility (79.5% to 93.5%) when encapsulated with GA/MD. Likewise, EOs from sweet lemon showed the highest phenolic content when using GA/CAS (228.27 mg GAE/g sample), and the encapsulated EOs of sweet lemon and mandarin with GA/MD/CAS (1709 and 1599 μmol TE/g) had higher antioxidant capacity. On the other hand, higher encapsulation efficiency was obtained in EOs of lime encapsulated with GA/MD (68.5%), and the nanoencapsulates of EOs from sweet lemon with GA/MD had higher D-limonene content (613 ng/mL). Using gum arabic and maltodextrin increased the encapsulation efficiency and D-limonene content in EO of sweet lemon. On the other hand, the formulations with casein were the most efficient wall materials for retaining D-limonene from the EOs of mandarin, lime, and orange.

## 1. Introduction

Nanoencapsulation has found increasing interest due to the demand for healthy foods or foods with a minimum of chemical additives [1] that allow to maintain the stability and availability of bioactive compounds [2,3,4]. Therefore, the use of nanoencapsulation of phenolic compounds, antioxidants, food colorants, antimicrobial agents, essential oils, minerals, flavorings, and vitamins is being studied to maintain or improve the foods’ properties, such as their sensory and nutritional attributes [5].

Natural essential oils (EOs) are a complex mixture of volatile compounds, mainly terpenes, terpenoids, and phenols [6,7] distributed in different parts of plants, including the fruit peel, as is the case of citrus fruits. Citrus EOs contain monoterpenes and sesquiterpenes, with the main component being D-limonene [8,9]. This compound has various uses in the food industry, highlighting its antioxidant, anti-inflammatory, antimicrobial, antiviral, and antifungal properties [3,8,10,11]. However, the potential of citrus EOs is affected by external factors (temperature, UV radiation, pH) [10] given their chemical nature (monoterpene hydrocarbons) [12], limiting their shelf life during food processing or storage [10,12,13].

Protecting EO compounds against this, and according to Hasani et al. [12], is necessary. Given this, a physical barrier using the nanoencapsulation technique promises to prevent the degradation of EOs and their bioactive components [7,10,14,15]. The use of nanoencapsulation in the food industry allows the controlled release of EOs, improving their bioavailability [6,10,13,14]. Moreover, it prolongs the shelf life of EOs by reducing their interaction with the food matrix and preventing their degradation or volatilization during storage [15,16,17,18], preserving their physical and oxidative properties over time [2,19].

Nanoencapsulation employs wall materials in the encapsulants [2], such as maltodextrin (MD) and gum arabic (GA) [20]. However, Mahdi et al. [15] and Moser et al. [21] mention that employing a single wall material does not ensure the ability to encapsulate efficiently, making EOs unstable [22,23], leading to the need to use mixtures with other compounds such as proteins, vitamins, carbohydrates, among others [23,24]. On the other hand, it is also necessary to pay attention to the encapsulation methods such as drying and lyophilization [25], the latter being more efficient for separating nanoparticles by expelling water without altering their structure and shape [5].

In this study, maltodextrin (MD), gum arabic (GA), and casein (CAS) were selected as wall materials due to their complementary properties that enhance the encapsulation efficiency and stability of essential oils [26,27]. MD is a low-viscosity polysaccharide with excellent solubility and film-forming capacity, making it suitable for processes such as spray drying and lyophilization, as well as being affordable and tasteless [15,20]. Gum arabic is a highly effective natural emulsifier due to its amphiphilic nature, stabilizing the oil–water interface and contributing to the controlled release of encapsulated compounds [21,23]. Casein is a milk-derived protein that strengthens the structure of the capsules through its hydrogen bonding and hydrophobic properties, which help to ensure the stability of compounds such as D-limonene against adverse environmental factors [24]. On the other hand, other materials such as sodium alginate, chitosan, gelatin, or vegetable proteins, although useful in some applications, have limitations such as high hydrophilicity, thermal sensitivity, or additional stabilization requirements that make them less suitable for this particular case [22,23,28]. In this way, the combination of MD, GA, and casein provides a synergistic approach that improves the encapsulation, stability, and functionality of the essential oils.

Given the importance of preserving key bioactive compounds such as D-limonene, the main component of citrus essential oils [29,30,31], the nanoencapsulation system is critical. The stability of D-limonene is often compromised due to its sensitivity to environmental factors such as oxidation and volatilization, which can lead to the loss of its functional and sensory properties [32]. Therefore, in order to maximize the preservation of essential oil properties and functionality, the selection of suitable wall materials and nanoencapsulation techniques is critical [4]. Nanoencapsulation protects the bioactive compounds of essential oils from external degradation and facilitates their controlled release [13,33]. It also allows for controlled delivery of bioactive compounds, enhancing their functional properties. Nanoencapsulation offers advantages over other encapsulation techniques, such as liposomal encapsulation or cyclodextrin-based complexes, by being more scalable, cost-affordable, and versatile in terms of the types of materials that can be used as carriers [34,35,36]. This makes nanoencapsulation a promising strategy to improve the shelf life and bioactivity of citrus essential oils [4,13], addressing the growing demand for natural and sustainable ingredients for the food industry [13]. Therefore, this study evaluated the efficiency of nanoencapsulation with different wall material formulations to preserve D-limonene and their impact on the physicochemical properties of nanoencapsulated essential oils, providing valuable information for practical application.

## 2. Materials and Methods

### 2.1. Materials

Orange (*Citrus sinensis*) fruits were obtained from the local market in the district of Chachapoyas (6°13′46′46″ S, 77°52′21″ W, 2338 m.a.s.l.). Lime (*Citrus limetta* Risso) and mandarin (*Citrus reticulata*) were obtained from the district of Bagua (5°38′21′21″ S, 78°31′ 54″ W, 408 m a.s.l.). Sweet lemon (*Citrus limetta* sp.) was obtained from the district of San Nicolás (6°23′45″ S, 77°28′56″ W, 1616 m.a.s.l.). All fruits in the trial were washed. The pulps were discarded, and the peels were cut into 1 cm^2^ pieces and stored separately in vacuum storage bags at 4 °C until use.

### 2.2. Chemical Reagents and Equipment

The following reagents and materials were used: Folin–Ciocalteu reagent, gallic acid, quercetin, sodium acetate (CH3COONa), aluminum chloride, 2,2-diphenyl-1-picrylhydrazyl (DPPH), acetonitrile, 2,2′-azino-bis-3-ethylbenzothiazoline-6-sulfonic acid (ABTS+), maltodextrin (MD) and gum arabic (GA), casein (CAS), ethyl acetate, n-alkanes standard (Supelco, Sigma Aldrich, St. Louis, MO, USA), potassium persulfate (Sigma Aldrich, St. Louis, MO, USA), sodium carbonate (Spectrum Chemical Mfg. Corp, New Brunswick, NJ, USA, 99.5%), and methanol (JT Baker, Deventer, The Netherlands).

The equipment used in this study was: steam distiller (Tecnal, TE-2761, Brazil), vortex (Analog Mini Vortex Mixers, OHAUS, Pine Brook, NJ, USA), freeze-drier (Labconco, 710402010 model, Kansas City, MO, USA), spectrophotometer (EMCLAB, EMC-11-UV Spectrophotometer, Duisburg, Germany), gas chromatograph (Agilent Technologies, 7890B GC System, Santa Clara, CA, USA), scanning electron microscope (ZEISS, V7.05, Jena, Germany), and FTIR spectrometer (Thermo Fisher Scientific Co., Nicolet Is50 FT-IR, Waltham, MA, USA).

### 2.3. Essential Oil Extraction

Essential oils from citrus peels were extracted separately by steam distillation (Tecnal, TE-2761, Piracicaba, Brazil) for 1 h, at an extraction temperature of 80 °C [31]. For this purpose, 300 g of peel in 500 mL of distilled water was used. After each extraction, an aqueous phase (aromatic water) and an organic phase (essential oil) were obtained and separated by decantation. The organic phase was stored at 2 °C until further use [37]. The EO yield was calculated as described by Prommaban and Chaiyana [38] using Equation (1).(1)$$Y\;\left(\%\right)=\left[\frac{A}{B}\right]\times 100$$
where: A is the volume of EO (mL), and B is the weight of the fruit peel (g).

### 2.4. Preparation of Nanoemulsions

Prior to preparation of the nanoemulsions, combinations of encapsulating walls were formulated using maltodextrin (MD), gum arabic (GA), and casein (CAS), as described in Table 1. The formulations followed Karaaslan et al. [20] because it provides antimicrobial activity and oxidative stability when employed in essential oils.

The method described by Karaaslan et al. [20] was applied considering a wall material ratio of 1/1 (*w*/*w*) and 1/1/1 (*w*/*w*/*w*). For this purpose, each encapsulating material was weighed according to the amount of essential oil to be encapsulated. Then, the wall materials were dissolved with 50 mL of ultrapure water for every 15 g solid material, and refrigerated at 5 °C for 12 h [39]. Then, the wall materials were mixed with the essential oil (30% dry weight) and homogenized in a vortex (Analog Mini Vortex Mixers, OHAUS, Pine Brook, NJ, USA). Finally, tween 80 (5% of the dry weight of the wall material) was added [40] and homogenized for 3 min until emulsification.

### 2.5. Freeze-Drying Process

The EOs nanoemulsions were frozen at −79 °C for 24 h, then freeze-dried (Labconco, 710402010 model, Kansas City, MO, USA) at 0.008 bar at −84 °C for 18 h. Subsequently, the material obtained was crushed in a mortar to form a homogeneous powder and placed in falcon tubes wrapped with aluminum foil to avoid light. Then, the samples (Appendix A, see in Appendix A) were stored in a polystyrene box avoiding humidity until their analysis [40,41].

### 2.6. Recovered Solids Yield

The yield of recovered powder (SY) was calculated using Equation (2) (Appendix A, see in Appendix A), following Di Giorgio et al. [42].(2)$$\mathrm{SY}\left(\%\right)=\frac{W_{f}}{W_{i}}100\%$$
where: SY is the yield of recovered solids (%), W_*i*_ is the initial weight of the wall materials (g, dry basis), and W_*f*_ is the weight of the powder obtained after the freeze-drying process (g, dry basis).

### 2.7. Moisture Content, Solubility, and Hygroscopicity of Nanoencapsulates

The moisture content was determined using a moisture analyzer (KERN Dab, Balingen, Germany). One gram of each nanoencapsulate was placed in the measuring pan and heated to 120 °C; the moisture content was recorded once a constant weight was reached [43].

The solubility percentage was determined following the method described by Ferraz et al. [44] with some modifications. For this purpose, 0.5 g of each powder was weighed, mixed with 25 mL of distilled water, and homogenized in a vortex at 80 rpm for 5 min. Subsequently, the solution was centrifuged at 5000 rpm for 10 min. Finally, a 25 mL aliquot of the supernatant was placed in a pre-weighed Petri dish and dehydrated at 105 °C for 12 h. The solubility of the powders was calculated using Equation (3).(3)$$\mathrm{Solubility}\;(\%)=\frac{\mathrm{Powder}\;\mathrm{weight}\;\mathrm{in}\;\mathrm{the}\;\mathrm{supernatant}}{\mathrm{Powder}\;\mathrm{weight}\;\mathrm{in}\;\mathrm{the}\;\mathrm{solution}}\;\times \;100$$

On the other hand, hygroscopicity was determined following the methodology described by Al-Maqtari et al. [17]. First, 0.5 g of nanoencapsulated essential oil was stored with a saturated sodium chloride solution at 25 °C and a relative humidity of 75.3%. After 1 week, the samples were weighed and expressed as a percentage of moisture absorbed, following Equation (4).(4)$$\mathrm{Hygroscopicity}\;(\%)=\frac{\mathrm{Absorbed}\;\mathrm{humidity}}{\mathrm{Sample}\;\mathrm{weight}}\;\times \;100$$

### 2.8. Encapsulation Efficiency

Encapsulation efficiency was determined through the content of phenolic compounds according to Radünz et al. [45]. For this purpose, the content of total phenolic compounds in un-encapsulated and nanoencapsulated EOs was measured by the Folin–Ciocalteu method. First, 0.5 mL of EOs was homogenized with 4.5 mL of methanol, 500 uL of the supernatant was separated, and 2.5 mL of diluted Folin plus 2 mL of 7.5% sodium carbonate were added. The solution was incubated for 2 h in the dark, and the absorbance was read (EMCLAB, EMC-11-UV Spectrophotometer, Duisburg, Germany) at 764 nm [11]. A standard curve prepared with gallic acid was used to quantify the phenolic compounds, and the results were expressed as mg gallic acid equivalent (GAE) per gram of sample. The analyses were performed in quadruplicate (Appendix A, see in Appendix A).

### 2.9. Total Phenolic Content

According to Papoutsis et al. [46], the total phenolic content of the aqueous extract of the nanoparticles of each essential oil was determined by the Folin–Ciocalteu colorimetric method. For this purpose, 20 mL of Folin–Ciocalteu reagent was diluted in 180 mL of water (diluted Folin solution), and 7.5 g of sodium carbonate was weighed and dissolved in 100 mL of water. To obtain the 1:10 aqueous extract, 0.5 g of sample was weighed and dissolved in 4.5 mL of methanol.

Then, it was homogenized in a vortex for 5 min and centrifuged at 5000 rpm for 30 min to obtain the supernatant, from which 500 µL was separated, 2.5 mL of the diluted Folin solution, and 2 mL of the 7.5% sodium carbonate solution were added. The mixture was placed for 2 h in the dark at room temperature. Finally, the absorbance of a blank (methanol) was read at 764 nm in a UV-Vis spectrophotometer (EMCLAB, EMC-11-UV Spectrophotometer). To quantify the total phenolic content, a curve (Y = 0.0004x + 0.0212) was developed using the gallic acid standard with concentrations between 0 and 2500 mM and a coefficient of determination (R^2^) of 0.9986. The results were expressed as mg of gallic acid equivalent per gram of sample (mg GAE/g sample). The analyses were performed in quadruplicate.

### 2.10. Stability of Antioxidant Activity by the DPPH Assay

The free radical scavenging activity of DPPH was determined following the method described by Al-Idee et al. [18] with some modifications. An aqueous extract was prepared, from which 1 mL of the supernatant was taken, and 1 mL of methanolic DPPH solution (0.1 mM) was added. The mixture was incubated in the dark for 30 min at room temperature, and the absorbance was read at 517 nm (EMCLAB, EMC-11-UV Spectrophotometer). Radical scavenging activity was calculated using Equation (5).(5)$$\mathrm{Inhibition}\;\left(\%\right)=\frac{\mathrm{Blank}\;\mathrm{absorbance}\;-\;\mathrm{Sample}\;\mathrm{absorbance}}{\mathrm{Blank}\;\mathrm{absorbance}}\;\times \;100$$

### 2.11. Stability of Antioxidant Capacity Using the ABTS+ Assay

With some modifications, the ABTS+ radical activity was assayed as described by Khalifa et al. [47] and Balcázar-Zumaeta et al. [48]. For ABTS+ (7.0 mM) solution, 88 µL of ABTS+ was mixed with 5 mL of potassium persulfate (2.45 mM) and incubated for 16 h at room temperature in the dark. To obtain an absorbance of 0.700 ± 0.002 at 734 nm, the solution was diluted with a volume of 200 to 300 mL of ethanol. Then 500 µL of each sample was added to 2.0 mL of ABTS+ solution, and the absorbance at 734 nm was measured against a blank. To determine the antioxidant activity, a standard curve was performed using Trolox (y = −0.0004x + 0.8502, R^2^ = 0.99). Finally, the results were expressed in µmol TE/g sample.

### 2.12. Characterization of Volatile Compounds by GC-MS

The sample extract was prepared by weighing 0.5 g of powder of each nanoencapsulated essential oil, and 4.5 mL of ethyl acetate was added, shaken, and placed in a centrifuge at 5000 rpm for 5 min. Then, the supernatant was separated, and 1 mL was filtered with a PTFE filter (0.45 µm). The analysis of the volatile compounds of the essential oil nanocapsules was performed by gas chromatography on a GC system (Agilent Technologies, 7890B GC System) coupled to a quadrupole mass spectrometer (MSD 5977B) equipped with a DB-5MS UI capillary column (60 m × 0.25 µm diameter × 1 µm thick). Helium was used as carrier gas at a 1 mL.min^−1^ flow rate.

The injection was performed in split mode (40:1), with an injected volume of 0.5 μL. The temperature of the mass detector, injector, and ionization source were 250 °C, 150 °C, and 280 °C, respectively. The oven temperature was maintained at 60 °C for 6 min, then increased to 270 °C at a rate of 3 °C min^−1^ and maintained at 270 °C for 4 min. The ionization voltage was 70 eV. Mass spectral data were obtained in scan mode over a mass range (*m*/*z*) of 45 to 450 amu. The detected compounds were identified by comparing the mass spectra with the National Institute of Standards and Technology (NIST Library 17) database. The identification of the compounds was confirmed by injecting the n-alkanes standard (C_8_–C_20_) and by comparing their retention indices [49] (Appendix A, see in Appendix A). The external standard of (R)-(+)- limonene and the nanocapsules of the essential oils were diluted in ethyl acetate (C_4_H_8_O_2_). The external standard was used at different concentrations (100, 50, 25, 12.5, 6.25 μg mL^−1^) to form the calibration curve and quantify the D-limonene content [11] (Appendix A, see in Appendix A).

### 2.13. Scanning Electron Microscopy (SEM)

High-resolution images of the essential oil nanocapsules were taken using a scanning electron microscopy (ZEISS, V7.05, Oberkochen, Germany) following the method described by Kim et al. [50]. The powder samples were placed on a double-sided black carbon tape mounted on a stainless steel pin. The samples were then coated with an ultra-thin layer of gold using a magnet sputter coater for 30 s in the metallizer (Q150RS Metalizador QUORUM, Quorum Technologies Ltd., Lewes, UK). Images of particle morphology and surface porosity/cross section were taken at 200×, 500× and 1000× magnification, respectively, with an accelerating voltage of 10 kV.

### 2.14. ATR FT-IR Spectroscopy

Spectral data were collected within the spectral range 4000–400 cm^−1^ using 32 scans and a spectral resolution of 4 cm^−1^ following the method of Kim et al. [50], Lei et al. [51], Timilsena et al. [52]. For this purpose, images were obtained using an FTIR spectrometer (Thermo Fisher Scientific Co., Nicolet Is50 FT-IR, Waltham, MA, USA) equipped with a single reflection diamond ATR sampling module.

### 2.15. Data Analysis

All data were subjected to an analysis of variance. Tukey’s multiple comparisons test was used to determine the possible significant differences (95%) between treatments, for which the statistical program RMarkdown (RStudio, version 2022.07.2 + 576, Boston, MA, USA) was used. The data were analyzed in quadruplicate.

## 3. Results and Discussion

### 3.1. Physical Properties of Essential Oil Nanocapsules

Solid yield

The yield of recovered solids is shown in Figure 1. The solid yields of the nanoencapsulated EOs of four citrus fruits showed significant differences according to the four wall materials used. Nanoparticles from sweet lemon, lime, and orange had the highest yield when using MD/CAS (Figure 1A–D) (96.4, 96.6, and 96.1%, respectively). The yields found were higher than those obtained by Mohammed et al. [53] and Mahdi et al. [54], who obtained yields between 77.61 and 90.22% when microencapsulating *Citrus aurantium* essential oil, and between 77.77 and 89.39% when microencapsulating cider extract with AG, WP, and MD. In our study, mandarin nanoparticles showed the highest yield when using GA/MD (Figure 1B) (95.8%). Mahdi et al. [54] found that nanocapsules with a high percentage of AG have a high powder yield due to their high film-forming ability. On the other hand, variations in the powder yield occur because of the high porosity generated by the formation of ice crystals. During freeze-drying, porosity is entirely eliminated to prevent structural collapse, with a minimal impact on the final volume [41,42].

Moisture content

Moisture content is a crucial characteristic of nanoparticles, indicating their quality and stability during storage [55]. Low content avoids caking problems, contributing to final acceptability [56]. In addition, low moisture is associated with efficient drying, bioactivity, oxidation, water activity, stickiness, and flowability. Hence, high moisture content affects storage stability as the wall material changes to a rubbery state, releasing and degrading the core material [57]. The nanoparticles of the four citrus fruits showed significant differences among the four types of wall materials employed (Table 2). It was observed that sweet lemon nanoparticles showed the lowest moisture content with GA/MD/CAS (4.57%). On the other hand, nanoparticles of mandarin and lime showed the lowest moisture content with MD/CAS (5.01% and 4.23%, respectively), and those of orange with GA/CAS (4.55%). The moisture content obtained was within the range reported by Mahdi et al. [15], with values between 3.87 and 5.71% for mandarin essential oil nanoencapsulated with whey protein (WPI), gum arabic (GA), and maltodextrin (MD). On the other hand, they are above 1.90 and 2.87%, as Mahdi et al. [57] reported when nanoencapsulating essential oil of mandarin with clove and cinnamon essential oils. Chew et al. [56] obtained moisture content percentages between 2.7 and 3.9% by microencapsulating kenaf seed oil using β-cyclodextrin/gum arabic/maltodextrin. The nanoparticles of EOs from lime, sweet lemon, and orange had less than 5% moisture content with the wall materials containing MD/CAS, GA/MD/CAS, and GA/CAS, respectively, indicating a prolonged storage capacity.

Solubility

Solubility is essential in determining the quality of wall materials used for the nanocapsules, since poorly soluble powders can cause problems during processing, resulting in economic losses [15]. The nanoparticles of the four citrus fruits show three groups with significant differences according to the wall materials, where the nanoparticles of sweet lemon, mandarin, lime, and orange reached the highest solubility with GA/MD (90%, 91.5%, 79.5%, and 93.5%, respectively) (Table 2). However, no significant differences in solubility were found between the four EOs nanoencapsulated using GA/CAS and MD/CAS.

The results obtained are in agreement with those reported by Rezende et al. [41], who found a higher solubility in extracts of bioactive compounds from the pulp and residues of acerola (*Malpighia emarginata* DC) microencapsulated with GA/MD, whose solubility was associated with the solubility of the wall materials used [58]. In studies on microencapsulation of ginger essential oil using MD and AG, a solubility of 84.6% was found [59]. However, this finding was lower than the highest solubility percentage (93.35%) found for the orange nanoparticles in our study, although they were within the values obtained by Karrar et al. [55], who reported a solubility between 86.61% and 90.90% when microencapsulating guru seed oil employing MD, AG, and WPI as wall materials. Solubility is important because it broadens the application possibilities of citrus peel essential oils in food products.

Hygroscopicity

The ability to absorb moisture, known as hygroscopicity, is crucial for evaluating the quality of nanoencapsulated oils. This characteristic is particularly significant, as water absorption during storage can lead to lipid oxidation [15], influencing the nutritional and flow characteristics of the powder [17]. The nanoparticles of sweet lemon showed significant differences among the four types of wall material used, with GA/MD giving the lowest hygroscopicity (6%). The encapsulating wall conditions is related to the hygroscopic capacity of the nanoencapsulant; thus, vast ranges can be observed, for sweet lemon and mandarin, with values from 6.3 to 16.5% and 7.1 to 16.3% (Table 2). However, the ranges observed in mandarin (7 to 16%) were within the values obtained by Mahdi et al. [15] for mandarin oil (12.05 and 15.42%) encapsulated with GA/MD/WPI. On the other hand, the hygroscopicity exhibited by the other citrus fruits was within the range reported by Karrar et al. [55] for microcapsules of guru seed oil encapsulated with MD, AG, and whey protein (6.95% and 8.76%). On their side, Locali et al. [59] reported values for pink pepper essential oil microcapsules (8.5% and 9.3%). On the other hand, formulations containing MD showed the lowest hygroscopicity since it is considered a low hygroscopicity wall material [44].

Encapsulation efficiency

The encapsulation efficiency of sweet lemon nanoparticles was affected according to the wall material used. For this citrus variety, the highest encapsulation efficiency was 75.5%. On the other hand, the encapsulation efficiency of mandarin nanoparticles did not show significant differences between the wall materials used, with an average efficiency of 68.9%. lime presented significant differences according to the encapsulating materials, having GA/MD the highest encapsulation efficiency (68.5%), while orange nanoparticles showed two significant heterogeneous groups with 29.5% and 41.9% (Table 2).

This result could be because gum arabic, a highly branched heteropolymer of sugars with a small amount of protein covalently bonded to the carbohydrate chain, is a film-forming agent [20]. This property allows it to entrap the encapsulated molecule effectively. Maltodextrin, a widely used glucose polymer, is known for its high solubility and low viscosity, making it an ideal coating material. The properties of these substances, such as the stretching vibration of the C-H bonds of alkenes, the stretching vibration of the C-H bonds of CH^2^, the stretching vibration of carbonyls (-C=O), the stretching vibration of aromatic rings, and the stretching vibration of the C-C bonds of glucose, contribute to their distinct applications in coating [60]. Therefore, these properties play a crucial role in essential oils’ emulsification and coating process.

The results are below those of Radünz et al. [45], who obtained an encapsulation efficiency (88.9%) of thyme essential oil when encapsulating maltodextrin and casein as wall materials. On the other hand, Mahdi et al. [54] obtained an efficiency between 72.11% and 87.20% when encapsulating citron extract with GA, modified starch, whey protein, and maltodextrin. In this study, the values were above from what Shetta et al. [61] obtained, reporting values between 8.15% and 23.1% when encapsulating peppermint and green tea essential oils with chitosan. The variations in the encapsulation efficiency (%) are due to the oil components’ affinity with the wall materials. The efficiency in encapsulating phenolic compounds is associated with the structure, ability to form films, and solubility of the wall material used [46]. In addition, Mahdi et al. [54] mention that the presence of whey protein and the absence of GA in formulations negatively affects encapsulation efficiency, which is congruent with what was found in this study, as GA/MD conferred the highest encapsulation efficiency to lime and orange nanoparticles.

The efficiency, solubility and hygroscopicity of citrus peel essential oils encapsulation is highly dependent on the physicochemical properties of the wall materials used. Previous studies have shown that combinations such as gum arabic (GA) and maltodextrin (MD) provide excellent film-forming capacity and solubility, allowing a more stable and efficient encapsulation of phenolic and antioxidant compounds [45,54,62,63,64]. Additionally, it safeguards sensitive compounds from oxidation and thermal degradation while permitting a controlled release of compounds, enhancing their bioavailability in food and pharmaceutical applications [61,65]. In contrast, materials such as chitosan show limited interaction with certain lipophilic compounds, explaining the lower encapsulation efficiencies observed in other studies [61]. Moreover, when utilizing protein sources like casein or whey protein, it’s crucial to include carbohydrates such as gum arabic. Otherwise, the effectiveness of these combinations may be compromised [54]. This behavior emphasizes the need for selecting wall material combinations that maximize both molecular compatibility and functional properties for specific applications [66,67]. Using appropriate wall materials, nanocapsules enhance the stability of essential oils’ chemical properties. This indicates their potential application in the food industry and the ability to achieve synergistic effects in the nanoencapsulation of bioactive ingredients [68,69].

The stable chemical properties that characterize nanocapsulation of essential oils make it a versatile tool for the food industry. Its applications are varied, ranging from the controlled release of bioactive compounds to extending shelf life, all while preserving organoleptic properties, reducing hydrophobicity, and controlling microbiological aspects of food. This versatility also makes it a promising option for developing functional foods, depending on the type of wall material used and its potential synergistic effects with the bioactive ingredients.

### 3.2. Chemical Properties of Essential Oil Nanocapsules

Total phenolic content

The wall material for encapsulating EOs from sweet lemon significantly affected the phenolic content, with GA/CAS having the highest one (228.27 mg GAE/g sample, Table 3). These results agree with those reported by Al-Maqtari et al. [17], who found that combining AG with a protein improves the encapsulation efficiency of phenolic compounds. On the other hand, mandarin nanoparticles showed no significant differences between the types of wall materials used, with an average phenolic content of 150 mg GAE/g of sample. Nanoparticles of lime also showed significant differences among the wall materials, with MD/CAS giving the highest phenolic content (98.40 mg GAE/g sample), a fact in agreement with that reported by Papoutsis et al. [46], who found high phenolic content for encapsulations of citrus by-product extracts with MD and soy protein. The results found in this work could be attributed to the ability of proteins to interact with various wall materials, including MD, as they form colloidal particles that encapsulate polyphenols [70]. Orange nanoparticles showed two heterogeneous groups with significant differences with values of 65.5 mg GAE/g sample and 85.8 mg GAE/g sample (Table 3). CAS stands out in sweet lemon and lime nanoparticles because there is no association between the protein used as wall material and the polyphenol extracts, which ensures that this material grants better conservation and protection [71].

Stability of antioxidant activity by the DPPH assay

Due to its hydrogen-donating capacity, the DPPH method generates an antioxidant profile encompassing reactivity towards aqueous radicals through radical quenching. Phenolic compounds with multiple hydroxyl groups exhibit higher free radical scavenging activity, especially against DPPH [40,72]. Nanoparticles of sweet lemon showed two homogeneous groups with significant differences with inhibition of 69.1% and 80.7%. Nanoparticles of mandarin and lime showed no significant differences between encapsulating wall materials, with 71.9% and 70.1%, respectively. In contrast, nanoparticles of orange showed three heterogeneous groups of wall materials with significant differences, being GA/MD the one that provided the highest inhibition (69.5%) (Table 3). The results obtained are related to the phenolic content in citrus nanoparticles because phenolic compounds are antioxidants [73] as they act as electron donors to unstable free radical molecules [57]. In this regard, the high antioxidant activity of sweet lemon nanoparticles may be associated with their high phenolic content.

On the other hand, the DPPH radical scavenging activity exhibited by mandarin was higher than that reported by Kamal et al. [74], who found that the essential oil from the peel of this citrus fruit showed a moderate radical scavenging activity of 24.08% compared to that obtained in this study (71.9%). In addition, GA/MD was the encapsulant material that provided the highest antioxidant activity to orange nanoparticles (Table 3). This behavior was due to the ability of MD to protect the essential oil [75], which results in agreement with that obtained by Moosavy et al. [76], who obtained high antioxidant activity by microencapsulating lemon essential oil using MD. However, Khalifa et al. [47] reported that the amount of polyphenols and antioxidant activity of mulberry microparticles depended on the wall material containing whey protein (WP), reporting better protection than MD and AG. These differences may be because MD and AG are more efficient in encapsulating essential oils. After all, the formulations containing these compounds had higher phenolic content and antioxidant capacity. The antioxidant activity of citrus essential oil is related to its chemical composition (D-limonene, *α*-pinene, *α*-terpinene) since other authors have already reported this activity in essential oils rich in monoterpenes (D-limonene, *α*-pinene). Moreover, the synergistic effect between the compounds of essential oils enhances their antioxidant activity [77].

Stability of antioxidant capacity by ABTS^+^ assay

The ABTS^+^ assay is used to determine radical scavenging effects. In contrast to an H-donor, it leads to the formation of ABTS^+^ and then to a decolorization of the solution at 734 nm. The decrease in absorbance due to antioxidant capacity reflects a free radical scavenging capacity. A compound’s reducing capacity significantly indicates its antioxidant potential [78]. The antioxidant capacity of sweet lemon nanoparticles exhibited significant differences according to the type of wall material. Using GA/MD/CAS, conferred the highest antioxidant capacity (1709 μmol TE/g).

On the other hand, significant differences were observed in the nanoparticles of mandarin, according to the wall material used. Encapsulating with GA/MD/CAS permitted the highest antioxidant capacity (1599 μmol TE/g). Likewise, the nanoparticles of lime exhibited two homogeneous groups with significant differences with values of 1099 μmol TE/g and 1219 μmol TE/g, while in the nanoparticles of orange, the antioxidant capacity varied according to the wall material, where GA/CAS was the one that gave the highest antioxidant capacity (1059 μmol TE/g) (Table 3).

Volatile compounds from nanocapsules of essential oils by GC-MS

Citrus essential oils comprise hydrocarbons, aldehydes, esters, ketones, and some miscellaneous compounds ranging from 20 to 60 compounds per Citrus EOs [11,31]. Among these are the volatile compounds constituting 85% to 99%, comprising monoterpenes (97%), sesquiterpenes, and sesquiterpenoids [11]. The chemical composition of citrus nanoparticles is shown in Figure 2, in which 5 (100%) components were identified in the nanoparticles of nanoencapsulated EOs from mandarin (Figure 2A, Appendix A, see in Appendix A). It was also found that GA/MD was the most efficient wall material since it conserved 4 components. On the other hand, 10 (100%) components were identified in mandarin nanoparticles (Figure 2B), with GA/CAS being the most efficient material since it conserved 8 components. Also, 8 (100%) components were identified in lime nanoencapsulate (Figure 2C), where GA/CAS was the most efficient wall material since it conserved 5 components. In contrast, in the orange nanoparticles, 5 (100%) components were identified (Figure 2D), and GA/MD/CAS was the most efficient wall material since it conserved the 5 components.

D-Limonene content in nanoencapsulated EOs of citrus peels

Figure 3 shows the content of nanoencapsulated D-limonene in the four citrus fruits. It is observed that the nanoparticles of sweet lemon showed significant differences depending on the wall material used. It was observed that when GA/MD was used, the highest amount of D-limonene (Figure 3A) (613 ng/mL) was found. On the other hand, the D-limonene content of mandarin, lime, and orange EOs nanoencapsulated showed significant differences according to the type of wall material (Figure 3B–D), with MD/CAS, GA/CAS and GA/MD/CAS being the most efficient (721 ng/mL, 536 ng/mL, and 521 ng/mL of D-limonene, respectively).

Additionally, the relative amounts were 81.37, 54.65, 60.65, and 77.23% for nanoencapsulated EOs from sweet lemon, mandarin, lime, and orange, respectively (Figure 2). The results obtained are lower than those of Kang et al. [79], who found 97.2% D-limonene in citrus essential oil nanoemulsions. Similarly, Dao et al. [80] reported that mandarin peels essential oil contains between 80.3% and 96.2%. On the other hand, the values in our study are higher than those found by Himed et al. [77], reporting 67.08% of D-limonene for encapsulated *Citrus limon*. In the same way, Smeriglio et al. [49] obtained 48.91% of D-limonene in *C. lumia* Risso. In this last study, the content of D-limonene varied from 32 to 98% in the genus Citrus, 68.98% in sweet oranges, and 47.56% in lemons. The wall material that was most efficient in retaining the volatile components of sweet lemon was GA/MD, possibly due to the ability of maltodextrin to retain volatile compounds in encapsulated systems [40].

Scanning electron microscopy (SEM) of nanoencapsulated EOs from citrus peels

Scanning electron microscopy micrographs were taken for nanocapsules of essential oil of *C. limetta* Risso (Figure 4) *C. sinensis* (Figure 5), *C. limetta sp*. (Figure 6), and *C. reticulata* (Figure 7) using different combinations of wall materials: MD/GA (A), GA/CAS (B), MD/CAS (C), and MD/GA/CAS (D). The images show remarkable differences in surface morphology and particle structure depending on the formulation used [57,81,82,83]. Nanocapsules with MD/GA (A) had spherical particles with a smooth surface and no pores or cracks, indicating effective encapsulation and structural stability. Capsules with GA/CAS (B) showed a more irregular morphology, with folds and roughness, which could be attributed to the different interaction between gum arabic and casein. On the other hand, formulations with MD/CAS (C) showed more compact and homogeneous particle structures, indicating a better interaction between maltodextrin and casein as wall materials. Finally, nanocapsules with MD/GA/CAS (D) had a more defined and geometrical surface, with well-formed particles, suggesting an optimal synergy between the three wall materials (maltodextrin, gum arabic, and casein), similar structures to those reported by Mahdi et al. [57].

These morphological differences are directly related to the encapsulation capacity and structural stability of the nanocapsules [84]. In particular, the MD/GA/CAS formulation seems to provide the highest efficiency in terms of homogeneous particle formation and protection of chemical and volatile compounds (Figure 2 and Figure 3, respectively), in line with the results of Mahdi et al. [57]. In addition, the roughness observed in some formulations could affect the controlled release of essential oils, highlighting the importance of carefully selecting wall materials according to the intended application.

ATR FT-IR Spectroscopy of nanoencapsulated EOs from citrus peels

The ATR-FTIR spectra of nanoencapsulated essential oil of *C. limetta* Risso (Figure 8A), *C. limetta* sp. (Figure 8B), *C. reticulata* (Figure 8C), and *C. sinensis* (Figure 8D), using different wall materials (GA/CAS, MD/CAS, MD/GA, MD/CAS/GA), revealed characteristic peaks confirming the interaction of the essential oils with the encapsulation materials. In the 4000–3000 cm^−1^ region, a prominent peak was observed at 3316 cm^−1^, associated with O-H stretching vibrations due to the hydrophilic components of the wall materials (GA, CAS, MD) [85,86,87].

In the 3000–2800 cm^−1^ region, bands around 2878 cm^−1^ were detected, related to the C–H stretching vibrations of the essential oils and the structure of the aromatic rings [88,89,90]. In the 1800–1500 cm^−1^ region, peaks such as those at 1711 cm^−1^, associated with the C=O group, and signals between 1483 and 1423 cm^−1^, indicated interactions between the bioactive compounds and the encapsulation materials. Likewise, in the 1500–1000 cm^−1^ region, the peak at 1138 cm^−1^ confirmed the presence of C–O–C structures characteristic of the carbohydrates in the wall materials, while peaks in the 939–756 cm^−1^ range reflect the specific molecular characteristics of the essential oils. In particular, combinations of wall materials, especially MD/CAS/GA (Figure 8D), showed sharper spectra, suggesting a higher efficiency in encapsulation and protection of the bioactive compounds. This may be attributed to synergistic interactions between simple polysaccharides (maltodextrin), proteins (casein), and complex polysaccharides (gum arabic). These differences show that the type of wall material significantly influences the retention and stabilization of functional compounds, optimizing their potential for applications in food products [91].

## 4. Conclusions

The results highlight the importance of the wall material role in the stability and functional properties of encapsulated essential oils. The combination of wall materials not only influenced the physical yield and moisture content but also played a pivotal role in retaining bioactive compounds, such as phenols and D-limonene, which are essential for the its antioxidant and sensory properties. Moreover, the ability of gum arabic and casein to better conserve phenolic compounds was demonstrated. Also, the synergistic effects observed among maltodextrin, gum arabic, and casein combinations (GA/MD/CAS) underline the importance of using multi-component encapsulating systems for achieving optimal performance. These findings suggest that the strategic selection and combination of wall materials can be tailored to specific essential oils and their intended applications in the food and pharmaceutical industries, offering a versatile approach to enhancing product stability and functionality.

The best physical yields were obtained with the mixture of maltodextrin plus casein (MD/CAS) as an encapsulating wall for essential oils of sweet lemon, lime, and orange. Furthermore, adding gum arabic (GA/MD/CAS) to this mixture resulted in nanoparticles with lower moisture content in the essential oil of sweet lemon. Low moisture content was also obtained when MD/CAS was used to encapsulate the essential oil of mandarin and lime and gum arabic plus casein (GA/CAS) for the essential oil of orange. When gum arabic and maltodextrin (GA/MD) were used as an encapsulant, the solubility of the four essential oils was increased, and the hygroscopicity of sweet lemon essential oil nanoparticles was reduced. Gum arabic and casein (GA/CAS) allowed the encapsulation of higher phenolic compounds from the essential oil of sweet lemon. However, when gum arabic was replaced by maltodextrin (MD/CAS), phenols from the essential oil of lime were better conserved. Additionally, formulations containing gum arabic and casein (GA/MD/CAS, GA/CAS) better conserved the essential oil of sweet lemon, mandarin, and orange.

Encapsulation efficiency depends on the encapsulating wall material. Using gum arabic and maltodextrin (GA/MD) increased the encapsulation efficiency of lime EO. Similarly, it encapsulated more D-limonene from the essential oil of sweet lemon. On the other hand, the formulations with casein (MD/CAS, GA/CAS, and GA/MD/CAS) were the most efficient wall materials for retaining D-limonene from the EOs of mandarin, lime, and orange. Finally, it is confirmed that the encapsulant material depends on the type of essential oil and the physicochemical properties to be conserved.

## Figures and Tables

**Figure 1 polymers-17-00348-f001:**
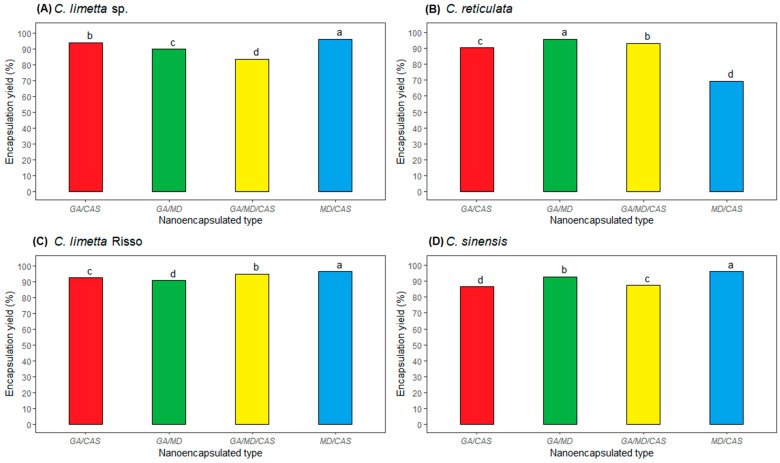
Solid yield of nanoparticles of (**A**) sweet lemon (*C*. *limetta* sp.), (**B**) mandarin (*C. reticulata*), (**C**) lime (*C. limetta* Risso), (**D**) orange (*C. sinensis*) EOs encapsulated with four wall materials.

**Figure 2 polymers-17-00348-f002:**
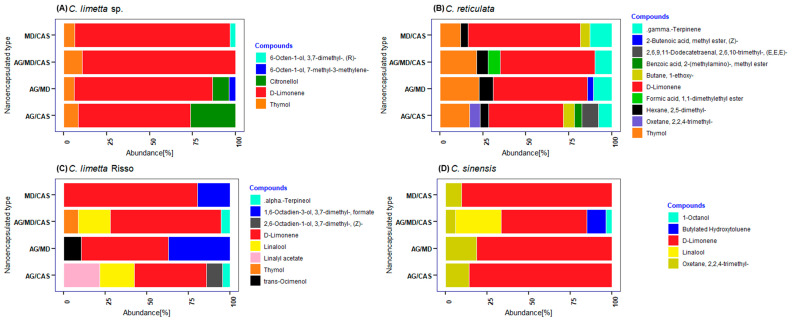
Chemical compounds found in (**A**) Sweet lemon (*C. limetta* sp.), (**B**) mandarin (*C. reticulata*), (**C**) lime (*C. limetta* Risso), and (**D**) orange (*C. sinensis*) EOs nanoencapsulated with four wall material.

**Figure 3 polymers-17-00348-f003:**
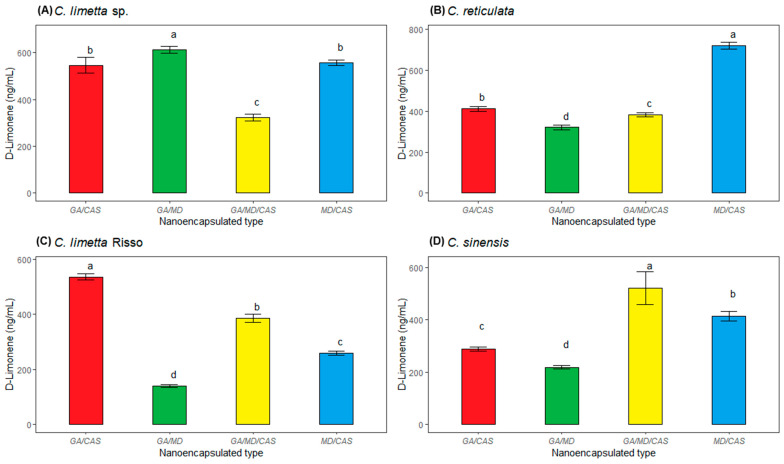
D-limonene concentration in (**A**) Sweet lemon (*C. limetta* sp.), (**B**) mandarin (*C. reticulata*), (**C**) lime (*C. limetta* Risso), and (**D**) orange (*C. sinensis*) EOs nanoencapsulated with four wall material. Different lower case letters in the graphs indicate significant differences (*p* ≤ 0.05).

**Figure 4 polymers-17-00348-f004:**
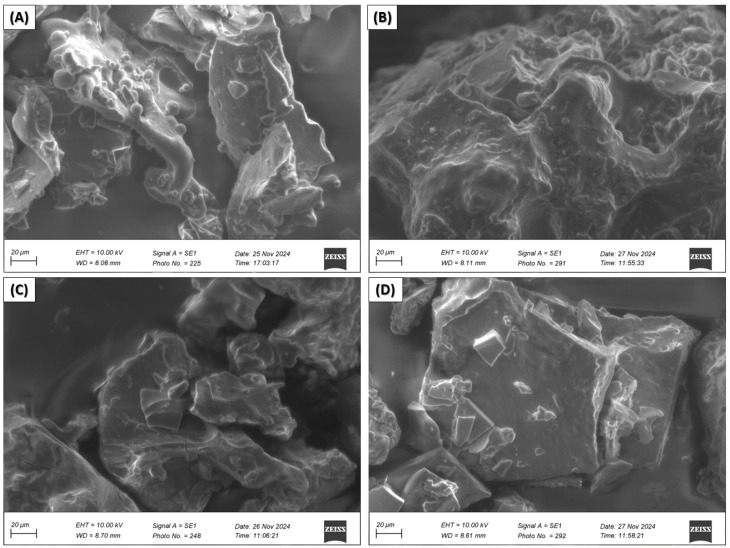
Scanning electron microscopy (SEM) of nanocapsules of EOs of *C. limetta* Risso with MD/GA (**A**), GA/CAS (**B**), MD/CAS (**C**), or MD/GA/CAS (**D**).

**Figure 5 polymers-17-00348-f005:**
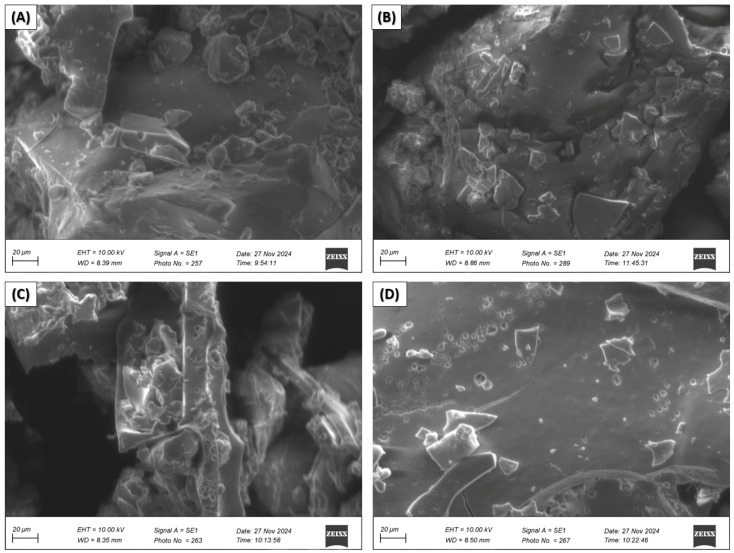
Scanning electron microscopy (SEM) of nanocapsules of EOs of *C. sinensis* with MD/GA (**A**), GA/CAS (**B**), MD/CAS (**C**), or MD/GA/CAS (**D**).

**Figure 6 polymers-17-00348-f006:**
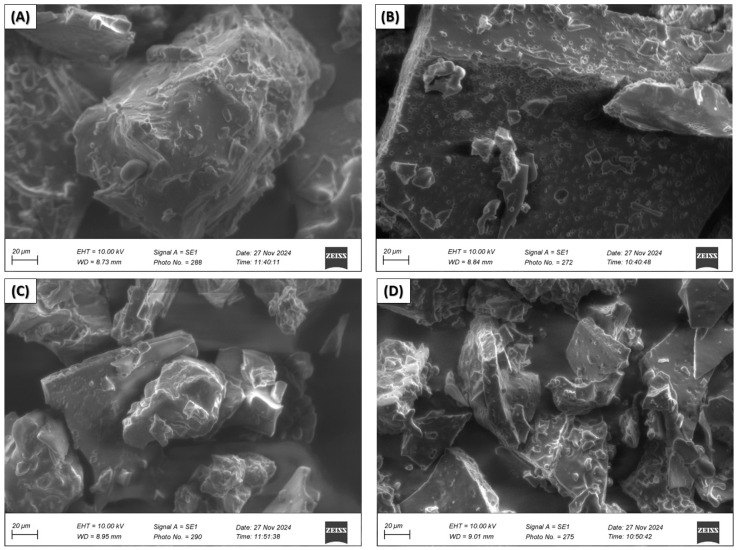
Scanning electron microscopy (SEM) of nanocapsules of EOs of *C. limetta* sp. with MD/GA (**A**), GA/CAS (**B**), MD/CAS (**C**), or MD/GA/CAS (**D**).

**Figure 7 polymers-17-00348-f007:**
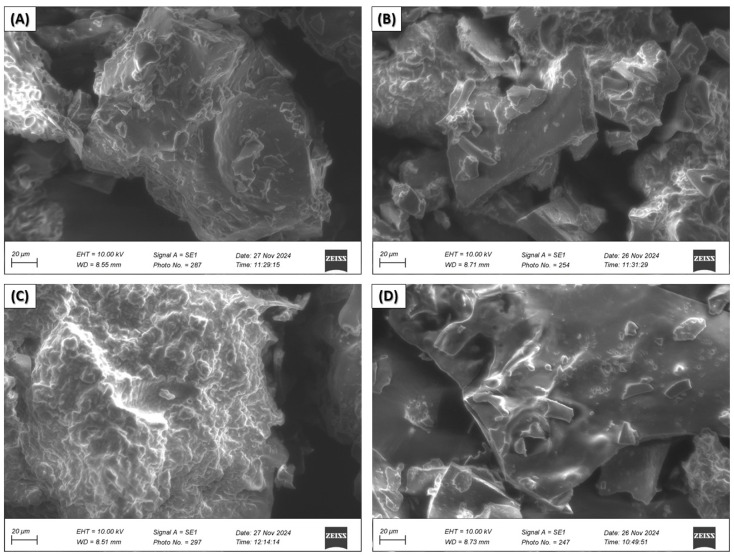
Scanning electron microscopy (SEM) of nanocapsules of EOs of *C. reticulata* with MD/GA (**A**), GA/CAS (**B**), MD/CAS (**C**) or MD/GA/CAS (**D**).

**Figure 8 polymers-17-00348-f008:**
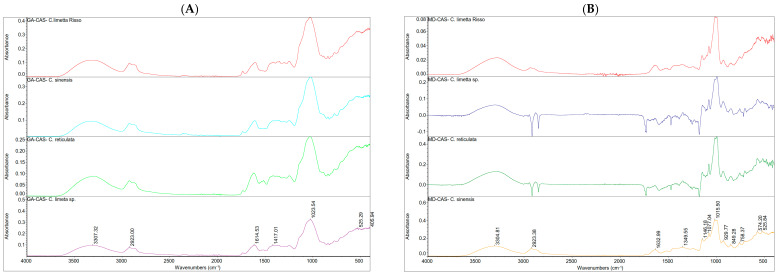
ATR-FT-IR spectra of nanoencapsulated essential oils of Sweet lemon (*C. limetta* sp.), mandarin (*C. reticulata*), lime (*C. limetta Risso*), and orange (*C. sinensis*) with four wall material: (**A**) GA/CAS, (**B**) MD/CAS, (**C**) MD/GA and (**D**) GA/MD/CAS.

**Table 1 polymers-17-00348-t001:** Nanoencapsulated formulation: EO/encapsulating wall ratio.

EO (mL)		Amount of Encapsulating Wall ^1^ (g)			
		MD/GA	MD/CAS	GA/CAS	GA/MD/CAS
Orange:	5	17	17	17	17
Mandarin:	7.25	24	24	24	24
Lime:	6.25	21	21	21	21
Sweet lime:	5	17	17	17	17

^1^ MD: maltodextrin; GA: gum arabic (GA); CAS: Casein. Wall material ratio of two components were 1/1 (*w*/*w*) and of three components were 1/1/1 (*w*/*w*/*w*). The total amount is presented in grams.

**Table 2 polymers-17-00348-t002:** Moisture, solubility, hygroscopicity, and encapsulation efficiency (EE) of four essential oils with different encapsulating walls.

Formulation		Moisture Content (%)	Solubility (%)	Hygroscopicity (%)	EE (%)
Sweet lemon	GA/MD	6.80 ± 00 ^a^	90 ± 3.27 ^a^	6 ± 30.55 ^d^	75.5 ± 0.98 ^a^
	GA/CAS	6.07 ± 00 ^b^	48 ± 1.63 ^c^	8.50 ± 1.00 ^c^	66.5 ± 1.04 ^c^
	MD/CAS	5.04 ± 00 ^c^	48 ± 7.11 ^c^	16.5 ± 1.00 ^a^	75.3 ± 0.50 ^a^
	GA/MD/CAS	4.57 ± 00 ^d^	63 ± 1.15 ^b^	12.5 ± 1.00 ^b^	69.8 ± 0.72 ^b^
Mandarin	GA/MD	6.06 ± 00 ^c^	91.5 ± 3.00 ^a^	7 ± 1.15 ^b^	68.9 ± 13.1 ^a^
	GA/CAS	7.36 ± 00 ^a^	78.5 ± 2.52 ^b^	8 ± 00 ^b^	65.2 ± 0.71 ^a^
	MD/CAS	5.01 ± 00 ^d^	74.5 ± 3.00 ^b^	16 ± 3.55 ^a^	65.4 ± 0.66 ^a^
	GA/MD/CAS	6.14 ± 00 ^b^	59.5 ± 1.00 ^c^	8.5 ± 1.00 ^b^	64.3 ± 1.00 ^a^
Lime	GA/MD	5.68 ± 00 ^b^	79.5 ± 12.5 ^a^	7.5 ± 1.00 ^b^	68.5 ± 0.95 ^a^
	GA/CAS	7.34 ± 00 ^a^	43.5 ± 1.91 ^c^	11 ± 1.15 ^a^	56.9 ± 1.59 ^bc^
	MD/CAS	4.23 ± 00 ^d^	45.5 ± 4.43 ^c^	6.5 ± 1.00 ^b^	54.8 ± 1.39 ^c^
	GA/MD/CAS	5.11 ± 00 ^c^	62.5 ± 4.43 ^b^	7 ± 1.15 ^b^	57.9 ± 1.64 ^b^
Orange	GA/MD	5.07 ± 00 ^b^	93.5 ± 1.00 ^a^	10 ± 00 ^a^	41.9 ± 4.27 ^a^
	GA/CAS	4.55 ± 00 ^d^	52.5 ± 1.00 ^c^	10 ± 00 ^a^	25.9 ± 5.25 ^b^
	MD/CAS	4.78 ± 00 ^c^	52 ± 7.11 ^c^	8 ± 3.55 ^b^	23.9 ± 2.00 ^b^
	GA/MD/CAS	6.24 ± 00 ^a^	61 ± 1.15 ^b^	7 ± 1.15 ^b^	29.5 ± 2.00 ^b^

The results are expressed as the mean ± SD. Different letters in the columns indicate significant differences (*p* ≤ 0.05) between the data.

**Table 3 polymers-17-00348-t003:** Total phenolic content (TPC), antioxidant capacity by DPPH assay, and by ABTS^+^ of nanoparticles of four essential oils with different types of encapsulants.

Formulation		TPC (mg GAE/g)	DPPH (% Inhibition)	ABTS^+^ (µmol TE/g)
Sweet lemon	GA/MD	167 ± 6.67 ^c^	80.7 ± 2.62 ^a^	1644 ± 35.8 ^b^
	GA/CAS	228 ± 7.08 ^a^	69.1 ± 1.66 ^b^	1537 ± 20.3 ^c^
	MD/CAS	168 ± 3.42 ^c^	67.2 ± 1.00 ^b^	1352 ± 6.29 ^d^
	GA/MD/CAS	205 ± 4.92 ^b^	79.3 ± 0.77 ^a^	1709 ± 18.2 ^a^
Mandarin	GA/MD	131 ± 55.0 ^a^	69.5 ± 1.66 ^a^	1255 ± 43.5 ^c^
	GA/CAS	146 ± 2.99 ^a^	69.0 ± 1.70 ^a^	1509 ± 25.8 ^b^
	MD/CAS	146 ± 2.78 ^a^	71.3 ± 1.25 ^a^	1512 ± 50.1 ^b^
	GA/MD/CAS	150 ± 4.21 ^a^	71.9 ± 1.83 ^a^	1599 ± 11.6 ^a^
Lime	GA/MD	68.6 ± 2.06 ^c^	70.1 ± 1.62 ^a^	1090 ± 10.3 ^b^
	GA/CAS	93.8 ± 3.47 ^ab^	68.1 ± 1.56 ^a^	1198 ± 16.8 ^a^
	MD/CAS	98.4 ± 3.03 ^a^	69.7 ± 1.80 ^a^	1219 ± 5.95 ^a^
	GA/MD/CAS	91.8 ± 3.57 ^b^	68.7 ± 1.82 ^a^	1099 ± 27.9 ^b^
Orange	GA/MD	65.5 ± 4.82 ^b^	69.5 ± 1.47 ^a^	1012 ± 25.7 ^b^
	GA/CAS	83.5 ± 5.92 ^a^	68.0 ± 0.45 ^ab^	1059 ± 13.6 ^a^
	MD/CAS	85.8 ± 2.25 ^a^	68.4 ± 1.92 ^ab^	1010 ± 27.5 ^b^
	GA/MD/CAS	79.5 ± 2.25 ^a^	66.5 ± 1.23 ^b^	1024 ± 9.68 ^ab^

The results are expressed as the mean ± SD. Different letters in the columns indicate significant differences (*p* ≤ 0.05) between the data.

## Data Availability

The original contributions presented in this study are included in the article/Appendix A.

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
