# Peer review of "Physicochemical Properties of Nanoencapsulated Essential Oils: Optimizing D-Limonene Preservation"

_polymers, 2025, doi:10.3390/polym17030348_

Round 1
Reviewer 1 Report
Comments and Suggestions for Authors
This article investigated the nanoencapsulation of four essential oils (sweet lemon, mandarin, lime, and orange) and their physicochemical properties, with a particular focus on assessing the impact of different wall materials (gum arabic, maltodextrin, and casein) on the antioxidant activity and retention of D-limonene. The study is practical and especially relevant for enhancing the stability and functionality of essential oils in the food sector, making it particularly suitable for the development of health foods or functional foods. However, the article still needs improvement in several areas to enhance its scientific rigor and readability.
1. The article is not innovative enough, and this type of research is already relatively mature. The article does not present significant innovations and lacks breakthrough thinking on existing encapsulation technologies or wall material selection. It is suggested to further highlight the practical application advantages or innovativeness of this study in the discussion section.
2. Lines 45-46, “antimicrobial” and “antibacterial” are the same concept. Please correct.
3. The introduction section simply states that composite walls are more effectively encapsulated than single walls, and does not explain why maltodextrin, gum arabic, and casein are used as walls and not others, such as sodium alginate, chitosan, gelatin, soy protein, etc., so please explain! In addition to the various encapsulation methods for essential oils, including nanoemulsions, liposomal encapsulation, polymer-based encapsulation, and cyclodextrin complexation techniques, why is the method of nanoencapsulation used? Please explain in the introduction.
4. Materials and reagents are listed only for the source of the peel; experimentally relevant reagents are not listed.
5. Standard deviation not labelled in Figure 1?
6. The particle size size and micro morphology of the nanocapsules of essential oils should be provided.
7. The effect of different wall materials on encapsulation efficiency was mentioned in the paper, but an in-depth exploration of the mechanism was lacking. It is recommended to reveal the specific mechanism of action of wall materials and essential oil components through molecular level or structural characterisation.
8. The data in the results section, although exhaustive, were not fully explored in the discussion of the meaning behind the data. For example, for results such as encapsulation efficiency and antioxidant activity, the advantages of individual wall combinations and their possible application scenarios need to be explained in more detail.
9. There is insufficient discussion on the potential for practical application of essential oils in food. The application scenario of encapsulated essential oils in food, their stability or how to maintain their activity during processing deserve further exploration. Stability of nanocapsules should be provided.
Author Response
Dear reviewer, thank you very much for your comments and suggestions. The following are the changes made to the manuscript.
The article is not innovative enough, and this type of research is already relatively mature. The article does not present significant innovations and lacks breakthrough thinking on existing encapsulation technologies or wall material selection. It is suggested to further highlight the practical application advantages or innovativeness of this study in the discussion section.
Response: We added the more information [L. 80-97, 359-368, 382-399, 400-406, 523-539, 548-554, 563-570, 573-586, 588-599]
Lines 45-46, “antimicrobial” and “antibacterial” are the same concept. Please correct.
Response: Sorry for the mistake. We removed the term “antibacterial” [L. 44-45]
The introduction section simply states that composite walls are more effectively encapsulated than single walls, and does not explain why maltodextrin, gum arabic, and casein are used as walls and not others, such as sodium alginate, chitosan, gelatin, soy protein, etc., so please explain!
Response: We added the explanation [L. 64-79]
In addition to the various encapsulation methods for essential oils, including nanoemulsions, liposomal encapsulation, polymer-based encapsulation, and cyclodextrin complexation techniques, why is the method of nanoencapsulation used? Please explain in the introduction.
Response: We added the explanation [L. 80-97]
Materials and reagents are listed only for the source of the peel; experimentally relevant reagents are not listed.
Response: Thank for the comment. We added information about materials and reagents [L. 107-114]
Standard deviation not labelled in Figure 1?
Response: Thank you for your observation, because the standard deviation is too low (<0.01), it is not possible to visualize the Figure 1, but we added the supplementary data (see to Table S1)
The particle size and micro morphology of the nanocapsules of essential oils should be provided.
Response: Thank for the comment, we added the Scanning electron microscope (SEM) images [L. 523-561]
The effect of different wall materials on encapsulation efficiency was mentioned in the paper, but an in-depth exploration of the mechanism was lacking. It is recommended to reveal the specific mechanism of action of wall materials and essential oil components through molecular level or structural characterization.
Response: We added the information [L. 359-368]
The data in the results section, although exhaustive, were not fully explored in the discussion of the meaning behind the data. For example, for results such as encapsulation efficiency and antioxidant activity, the advantages of individual wall combinations and their possible application scenarios need to be explained in more detail.
Response: We added the more information [L. 359-368, 382-399, 548-554]
There is insufficient discussion on the potential for practical application of essential oils in food. The application scenario of encapsulated essential oils in food, their stability or how to maintain their activity during processing deserve further exploration. Stability of nanocapsules should be provided.
Response: We added the information [L. 400-406]

Reviewer 2 Report
Comments and Suggestions for Authors
Journal Title: Polymers
Manuscript Title: Nanoencapsulated essential oils physicochemical properties: optimizing D-limonene conservation
Manuscript ID: polymers-3259279-peer-review-v1
Authors: César R. Balcázar-Zumaeta et al.
This paper investigates the effectiveness of different wall material formulations in nanoencapsulating essential oils (EOs) from citrus peels to improve their antioxidant stability and conserve D-limonene content. By comparing combinations of gum arabic, maltodextrin, and casein, the study assesses how these materials impact the solubility, phenolic content, antioxidant capacity, and encapsulation efficiency of EOs from sweet lemon, mandarin, lime, and orange.
-The title could be improved for clarity and precision. Here’s the proposed version, clarifying that the focus is on studying the physicochemical properties of nanoencapsulated essential oils specifically to optimize the preservation (instead of "conservation") of D-limonene: "Physicochemical Properties of Nanoencapsulated Essential Oils: Optimizing D-Limonene Preservation"
-page 1, lines 17,18: the ambiguity of the phrase "A Essential oils exhibit antioxidant properties; however, they are prone to oxidative degradation due to environmental conditions, so it is essential to safeguard them." could be refined for clarity as follows: "Essential oils exhibit antioxidant properties but are prone to oxidative degradation under environmental conditions, making their preservation crucial."
-section 2.2. Essential oil extraction: please provide the working temperature, and if a vacuum was applied is should be stated
-page 2, line 84: the units of volume and weight should be mentioned
-section 2.1. Materials and reagents: all the reagents should be listed in the section (MD, AG,) CAS, tween 80, sodium chloride, Folin-Ciocalteu reagent, sodium carbonate, gallic acid, methanolic DPPH solution, ABTS+, potassium persulfate, methanol, ethanol, ethyl acetate
-section 2.3. Preparation of nanoemulsions: the rationale for choosing the amounts listed in Table 1 should be explained
-Table 1: in column one, the amount of mL of EOs is missing, please fix
-(Equation 3): edit "supernatan"
-(Equation 4): edit "Absroved"
-A section to describe all the equipment used in the study should be introduced for consistency
-(Equation 5): edit "absorvance"
-page 4, line 168: edit "88 uL"
-section 3.1. Physical properties of essential oil nanocapsules: the discussion must be improved, and additional characterization is needed to sustain the assumptions stated in between the lines 215-219 (page 5). I suggest the following methods that could effectively verify and characterize the morphology and porosity of the nanocapsules after freeze-drying: SEM, TEM, X-ray diffraction. In my opinion, SEM is a must!
-additionally, moisture, solubility, hygroscopicity, and encapsulation efficiency (EE) are to be described in relation to the envisaged structure of the respective nano encapsulated EO in their host matrices. At least FTIR spectroscopy should be provided to introspect the kind of possible interaction between the functional groups of the components. The authors need to find and discuss thus, mechanisms of interaction responsible for the behavior of their materials with respect to moisture, solubility, hygroscopicity, and encapsulation efficiency. The authors keep comparing with other studies but they need to explain WHY their products behave as such
-page 11, line 393: please revise the sentence "Citrus essential oils comprise hydrocarbons, aldehydes, esters, ketones, and 20 to 60 compounds."
-Figure 2: it is recommended to keep the color code for the same ingredient, at least for those with a higher abundance (D-Limonene)
-also in Fig 3, keep the color code for each category of nanoencapsulation type
Author Response
Dear reviewer, thank you very much for your comments and suggestions. The following are
the changes made to the manuscript.
The title could be improved for clarity and precision. Here’s the proposed version, clarifying that the focus is on studying the physicochemical properties of nanoencapsulated essential oils specifically to optimize the preservation (instead of "conservation") of D-limonene: "Physicochemical Properties of Nanoencapsulated Essential Oils: Optimizing D-Limonene Preservation"
Response: Thank for the suggestion, we changed the title [L. 2-3]
page 1, lines 17,18: the ambiguity of the phrase "A Essential oils exhibit antioxidant properties; however, they are prone to oxidative degradation due to environmental conditions, so it is essential to safeguard them." could be refined for clarity as follows: "Essential oils exhibit antioxidant properties but are prone to oxidative degradation under environmental conditions, making their preservation crucial."
Response: Thank for the comment, we clarified [L. 17-18]
section 2.2. Essential oil extraction: please provide the working temperature, and if a vacuum was applied is should be stated
Response: Thank you for the comment. We added the working temperature, and vacuum was not applied in the extraction [L. 123]
page 2, line 84: the units of volume and weight should be mentioned
Response: thank for the suggestion. We added required information [L. 129]
section 2.1. Materials and reagents: all the reagents should be listed in the section (MD, AG,) CAS, tween 80, sodium chloride, Folin-Ciocalteu reagent, sodium carbonate, gallic acid, methanolic DPPH solution, ABTS+, potassium persulfate, methanol, ethanol, ethyl acetate
Response: Thank for the comment. We added information about materials and reagents [L. 107-114]
section 2.3. Preparation of nanoemulsions: the rationale for choosing the amounts listed in Table 1 should be explained.
Response: Thank for the suggestion. We clarified the table [Table 1, L. 131-139]
Table 1: in column one, the amount of mL of EOs is missing, please fix
Response: Thank for the suggestion. We clarified the table [Table 1, L. 131-139]
(Equation 3): edit "supernatan"
Response: Sorry for the mistake. We changed for “supernatant” [L. 169]
(Equation 4): edit "Absroved"
Response: Sorry for the mistake. We changed for “absorbed” [L. 175]
A section to describe all the equipment used in the study should be introduced for consistency
Response: Thank for the comment. We added information about equipment used [L. 115-120]
(Equation 5): edit "absorvance"
Response: Sorry for the mistake. We changed for “absorbance” [L. 212]
page 4, line 168: edit "88 uL"
Response: Sorry for the mistake. We changed for “µL” [L. 215, 218]
section 3.1. Physical properties of essential oil nanocapsules: the discussion must be improved, and additional characterization is needed to sustain the assumptions stated in between the lines 215-219 (page 5). I suggest the following methods that could effectively verify and characterize the morphology and porosity of the nanocapsules after freeze-drying: SEM, TEM, X-ray diffraction. In my opinion, SEM is a must!
Response: Thank for the suggestion. We added SEM analysis in the study [L. 245-252, 523- 539, 548-554, Figure 4 – (541-543), Figure 5 – (544-546), Figure 6 – (556-558), and Figure 7 – (559-561)]
additionally, moisture, solubility, hygroscopicity, and encapsulation efficiency (EE) are to be described in relation to the envisaged structure of the respective nano encapsulated EO in their host matrices. At least FTIR spectroscopy should be provided to introspect the kind of possible interaction between the functional groups of the components. The authors need to find and discuss thus, mechanisms of interaction responsible for the behavior of their materials with respect to moisture, solubility, hygroscopicity, and encapsulation efficiency. The authors keep comparing with other studies but they need to explain WHY their products behave as such
Response: Thank for the suggestion. We added more information please see lines 359-368 and 382-399, FT-IR spectroscopy please see lines 563-586 and the Figure 8.
page 11, line 393: please revise the sentence "Citrus essential oils comprise hydrocarbons, aldehydes, esters, ketones, and 20 to 60 compounds."
Response: Thank for the suggestion. We rewrote the sentence and added the references [L. 481-484]
Figure 2: it is recommended to keep the color code for the same ingredient, at least for those with a higher abundance (D-Limonene)
Response: Thank for the suggestion. We modified the color in the figure for the compounds as D-Limonene, thymol and linalool (higher abundance) [L. 517-518]
also in Fig 3, keep the color code for each category of nanoencapsulation type
Response: Thank for the comment. We kept the color code for each category of nanoencapsulation type in the Figure 3 [L. 520-522], just like the Figure 1 [L. 281-283]
Round 2
Reviewer 1 Report
Comments and Suggestions for Authors
The ms. has improved a lot after revision, so I recommend its acceptance.